# Modulation pattern recognition method of wireless communication automatic system based on IABLN algorithm in intelligent system

**Ting Xie**[1], **Xing Han**[2] *

**1** Railway Department, Hohhot Vocational College, Hohhot, China, **2** School of Resources and Environment, Inner Mongolia University of Technology, Hohhot, China

* 15648131154@163.com

**Data Availability Statement:** All relevant data are within the manuscript and its Supporting Information files.

**Funding:** The author(s) received no specific funding for this work.

## Abstract

The aim of this study is to address the limitations of convolutional networks in recognizing modulation patterns. These networks are unable to utilize temporal information effectively for feature extraction and modulation pattern recognition, resulting in inefficient modulation pattern recognition. To address this issue, a signal modulation recognition method based on a two-way interactive temporal attention network algorithm has been developed. A two-way interactive temporal network is designed on the basis of the long and short-term memory network with the objective of enhancing the contextual connection of the temporal network. The output of the temporal network is attentively weighted using the soft attention mechanism. The proposed algorithm exhibited enhanced overall, average, and maximum recognition rates at varying signal-to-noise ratios, with an increase of 10.34%, 8.33%, and 3.33%, respectively, in comparison to other algorithms within the Radio Machine Learning (RML) 2016.10b dataset. Furthermore, the modulated signal recognition accuracy was as high as 92.84%, with an average increase in the Kappa coefficient of 12.28%. The Kappa coefficient in the Communication Signal Processing Benchmark for Machine Learning (CSPB. ML2018) 2018 dataset was 0.62, representing an average increase of 10.32% over other algorithms. The results demonstrate that the proposed recognition method can enhance the network's accuracy in recognizing modulated signals. Moreover, it has potential applications in modulation pattern recognition in automatic systems for wireless communications.

## 1. Introduction

As science and technology develop, wireless communication technology is constantly applied in daily life, facilitating people's material and cultural life [1]. Among the aforementioned fields, automatic modulation pattern recognition technology represents a significant area of wireless communication automation [2]. This technology is capable of automatically recognizing the modulation mode of wireless communication signals, which plays a pivotal role in

**Competing interests:** The authors have declared that no competing interests exist.

determining the radio's ability to perceive the spectrum space. However, with the exponential growth of communication data, the allocation and utilization of the limited spectrum resources has become a pressing concern in the current wireless communication field [3]. Traditional modulation pattern recognition methods are mainly achieved through likelihood ratio or feature extraction, and their computational steps are complex and the recognition accuracy is poor [4]. The advent of deep learning techniques thus opens up new avenues for modulation pattern recognition. The application of deep learning techniques to modulation pattern recognition has the potential to enhance the accuracy of recognition based on feature extraction, as evidenced by studies [5,6]. However, in the modulation pattern recognition of wireless communication, it is difficult for the algorithm formed by Convolutional Neural Network (CNN) to utilize the time series information. Meanwhile, recurrent networks are slow for long sequence training. Accordingly, the study proposes a two-way Interactive Attention Bi-LSTM Network (IABLN) algorithm with the objective of resolving the aforementioned issue. The number of chain cells in LSTM is reduced by enhancing the contextual linking ability of Long Short-Term Memory (LSTM) networks and losslessly compressing the length of information using convolutional networks. Finally, the attention mechanism weights the LSTM, and the IABLN algorithm is constructed to improve modulation pattern recognition and classification effectiveness for the automatic wireless communication system.

A two-way interactive temporal network based on LSTM is studied and designed to enhance the contextualization capability of the temporal network. By introducing multiple rounds of interactive operations, the ability to extract modulation pattern information for wireless communication systems is improved. The application of soft attention mechanisms within the IABLSTM network enables the model to prioritize pertinent information pertinent to the current task, thereby enhancing its overall performance. The modulation pattern recognition method for wireless communication automation systems, which is based on the IABLN algorithm developed in the study, addresses the limitations of temporal networks in long input sequences and the lack of temporal information sensitivity in convolutional networks. Overall, the recognition method improves the contextualization capability of the temporal network, speeds up the inference process, and improves the recognition accuracy of the network, which has a good application prospect in the field of wireless communication. The proposed recognition method has the potential to enhance the capabilities of temporal networks in summarizing past and future information, thereby facilitating the advancement of modulation pattern recognition technology for automatic systems. Furthermore, it offers theoretical and technical support for the development of intelligent communications.

The overall framework of the study can be divided into five sections. In Section 1, a synthesis is presented of the domestic and international achievements and shortcomings pertaining to modulation pattern recognition methods for the development of automatic wireless communication systems. In Section 2, the study proposes the IABLN algorithm and based on this, the design of modulation pattern recognition methods based on the IABLN algorithm is carried out. In Section 3, the experimental simulation and analysis are carried out. In the fourth part, the experimental results are discussed and analyzed. In Section 4, the research findings are summarized and directions for further research are pointed out.

## 2. Related works

Communication development promotes the improvement and enhancement of people's living standards, and modulation pattern recognition technology is significant in wireless communication. Scholars have achieved many in research on modulation pattern recognition of wireless communication systems. F. Liu et al. proposed a method using feature extraction and deep

learning for low accuracy of automatic recognition of wireless communication signals. By optimizing the gated recursive unit, the CNN and the parallel gated recursive unit were input for identification, achieving a high recognition rate with low Signal-to-noise Ratio (SNR) [7]. To improve automatic modulation classification in the development of cognitive radio, Q. Zheng et al. proposed a two-level data enhancement method using spectrum interference. By converting the original signal to the frequency domain, the frequency domain information was used to enhance the radio signal to help modulation classification [8]. To improve multi-class modulation modes classification of modulated signals, R. Khan et al. increased the modulated signals in the frequency and spatial domains by deploying three data enhancement methods: stochastic method/reduction, random shift and random weak Gaussian ambiguity enhancement techniques. Hyper-parameter selection based on cross-validation was used for statistics, and it was found that the learning efficiency in the spatial domain was better than that in the frequency domain [9]. To strengthen the analysis of the differences and characteristics of in-phase/quadrature and amplitude/phase representation, S. Chang et al. introduced CNN and Recursive Neural Network (RNN) into automatic modulation recognition for modulation type recognition of received signals. Through the proposed network, the effective use of all outputs was realized [10]. To improve low effectiveness of modulation classification in working systems with low SNR, Y. Sun et al. proposed a modulation type classification model for different received SNRs using machine learning. Firstly, the constellation image and image classification technology were used for modulation type detection. Secondly, the feature graphic representation was used to represent the statistical features as a spider map of machine learning. Therefore, the overall classification accuracy of 59.00% was obtained at 0dB SNR [11]. N. Rashvand et al. proposed a natural language processing Transformer network-based modulation recognition method with the objective of enhancing the accuracy of automatic modulation recognition of input signals. The RF signals were embedded with markers through real-time edge computing of IoT devices, resulting in an accuracy of 65.75% in the RML2016 dataset [12]. H. S. Ghanem et al. proposed a CNN-based modulation classification algorithm to address the problem of low modulation recognition efficiency due to point deformation and dispersion of constellation maps caused by noisy channels. The generation of modulation type travel graphs and the reliance on Radon variations for training and testing enabled the achievement of effective modulation classification under fading channel conditions [13].

Deep learning technologies such as CNN and LSTM are widely used in modulation classification in the field of wireless communication, and they have strong potential for processing and analyzing large data by obtaining raw data and finding representations for different tasks such as classification and detection. Therefore, deep learning technology's value in the modulation classification of intelligent communication systems has attracted many researchers. To improve gradient vanishing problem in the process of modulation pattern recognition, J. N. Njoku et al. proposed a cost-effective hybrid neural network c. The pooling layer, small filter size, Gaussian drop layer, and skipping connection layer were used to increase network capacity, so as to enhance its feature extraction process. Moreover, the recognition accuracy of 93.50% was achieved in the Deep-Sig dataset [14]. To improve that the modulated radio signal had a great influence on the modulation recognition results, S. Lin et al. proposed a time-frequency attention mechanism based on CNN, which solved the problem of learning channel, frequency, and time information [15]. To promote deep learning in radio signal recognition, Y. Tu et al. created a real-world radio signal dataset, and used deep learning methods and machine learning methods to compare the recognition benchmarks, so as to realize the automatic collection and labeling of data [16]. The computational speed of deep neural networks also has limitations. Ashtiani et al. proposed an integrated deep neural network. Sub-nanosecond image classification was performed by directly processing the light waves propagating

across the chip's pixel array, eliminating the need for large memory modules [17]. To improve the classification accuracy of higher-order modulations in the polar plane, A. H. Shah et al. proposed to use phase ordinariness and polarity as combined inputs to the feature proposal in CNNs, and to divide the CNN into four blocks, each consisting of a set of symmetric and asymmetric filters. The extended input of the network was improved by adding features inside the network, thus improving the network [18]. M. Venkatramanan and M. Chinnadurai developed a deep learning arithmetic optimization algorithm based on an augmented modulation classification method to address the suboptimal efficiency of multiple input multiple output orthogonal frequency division multiplexing systems. Modulated classification by LSTM-based CNN and hyperparameter selection of LSTM-CNN using enhanced modulation led to rational validation results in simulation tests [19].

Based on the above, it can be seen that scholars have carried out various studies on modulation pattern recognition of wireless communication systems using deep learning, and most of them focus on CNNs or temporal recurrent neural networks. However, the lack of global information and temporal characteristics of CNNs leads to the impoverishment of the contextual connection of modulated signals. In addition, the characteristics of serial computation in time series networks lead to certain limitations in the training scale and inefficient thrust efficiency. These limitations continue to impede the advancement of deep learning-based modulation patterns for wireless communication systems. To address this challenge, the study proposes a modulation pattern recognition method based on the IABLN algorithm. While enhancing LSTM context connection, the attention mechanism innovatively carries out the attention weighting of network output, and an IABLN bidirectional interaction algorithm is designed to improve modulation pattern recognition. In the field of wireless communications, the temporal characteristics of modulated signals are key to identifying different modulation patterns. The IABLN algorithm is designed based on these temporal characteristics to ensure that the different modulation patterns can be effectively captured and distinguished.

## 3. Design of modulation pattern recognition method using IABLN algorithm

Firstly, the contextual connection ability of LSTM is enhanced. Secondly, the length of lossless compressed information of a convolutional network is employed to reduce the number of chain cells of LSTM. Thirdly, a Bidirectional LSTM (BiLSTM) is introduced to construct an Interactive Bidirectional Timing Sequence Network (IBLSTM). Secondly, an attention mechanism is introduced to weight the output attention of IBLSTM, thereby forming the IABLN algorithm.

### 3.1. LSTM-based IBLSTM network

Automatic modulation recognition in intelligent communications represents a crucial aspect of recognizing the radio's capacity to perceive the spectral space. This is a fundamental and universal aspect of several fields. The current common method for automatic modulation pattern recognition is information modulation recognition using deep neural networks. However, modulation recognition based on convolutional networks lacks global information and temporal features, which can result in a weak contextual connection to the signal. Furthermore, temporal networks exhibit serial computation characteristics, which results in slower speeds for both large-scale training and inference. This results in a sub-optimal modulation recognition pattern for deep neural networks based on convolutional networks. Consequently, research is conducted to investigate alternative modulation pattern recognition methods. To improve the ability to extract modulation mode information from wireless communication systems, the

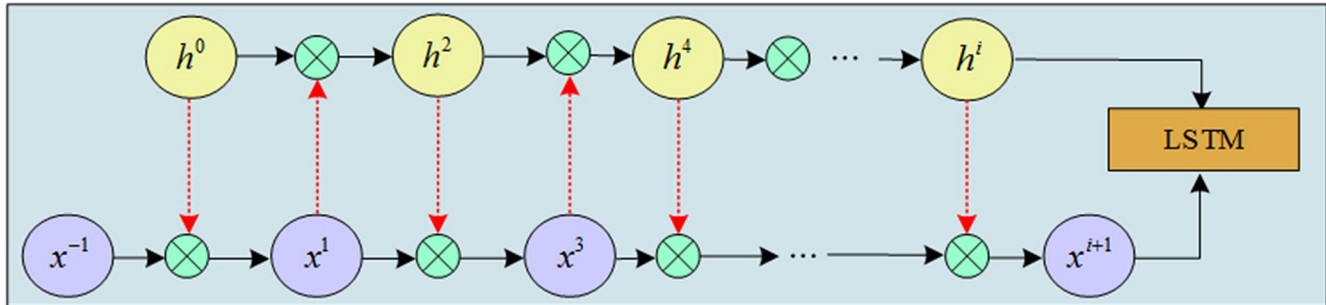

**Fig 1. Schematic diagram of multi-round interaction.**

context connection ability of LSTM is optimized. Throughout the execution process of the LSTM, input elements and hidden state intelligence from the previous time fragment interact within the LSTM, which leads to insufficient representation of the context [20,21]. Therefore, a deformed LSTM is introduced to enhance the context connection. Before input $x^i$ at the current moment enters the cell of the LSTM, $x^i$ is interacted with another input $h_{t-1}$ of the LSTM for multiple rounds, thus enhancing the contextual modeling capability. The specific expression is shown in Eq (1).

$$\begin{cases} x^i = 2\sigma(Q^i h_{prev}^{i-1}) \odot x^{i-2} \text{ for } odd \ i \in [1\ldots r] \\ h_{prev}^i = 2\sigma(R^i x^{i-1}) \odot h_{prev}^{i-2} \text{ for } even \ i \in [1\ldots r] \end{cases} \tag{1}$$

In Eq (1), $r$ represents the number of interactions. $\sigma$ represents the Sigmoid activation function. $Q^i$ and $R^i$ represent the parameters, respectively. $h_{prev}^i$ represents another input obtained after the interaction. Among them, $x_{-1} = x_t$, $h_{prev}^0 = h_{t-1}$, and both denote the current moment's input and the previous moment's output hidden state, respectively. A schematic diagram of the interaction of the input with another input of the LSTM for multiple rounds is shown in Fig 1.

Fig 1 shows two inputs interacting in 5 rounds. The entire stage is considered interactive, i.e., an interaction between the input vector and the hidden state. The LSTM can be interacted to encode sequence information from the current moment to the future moment [22,23]. However, it is not possible to encode information from the future moment to the current moment. Therefore, Bi-LSTM is further introduced for optimization. BiLSTM is a network model with forward LSTM and backward LSTM, better capturing the dependencies between long distances and realizing back-to-front information encoding, with superior global synthesis capabilities [24,25]. Fig 2 shows the BiLSTM network structure.

In Fig 2, the BiLSTM has a two-layer LSTM structure, the first layer is the forward LSTM, which is responsible for propagating the input information sequence forward. The second layer is the backward LSTM, which is responsible for propagating the input information backwards. The expression for the output of the BiLSTM at a certain point in time is shown in Eq (2) [26].

$$\begin{cases} h_t^F = f_{ac}(\mathbf{U}^F h_{t-1}^F + \mathbf{W}^F x_t + b^F) \\ h_t^B = f_{ac}(\mathbf{U}^B h_{t-1}^B + \mathbf{W}^B x_t + b^B) \\ h_t = h_t^F \oplus h_t^B \end{cases} \tag{2}$$

In Eq (2), $h_t^F$ is forward LSTM output, $h_t^B$ is backward LSTM output, $f_{ac}(\bullet)$ is the activation function, $\mathbf{U}$ and $\mathbf{W}$ are the weight matrices. $F$ and $B$ are the forward and backward LSTM,

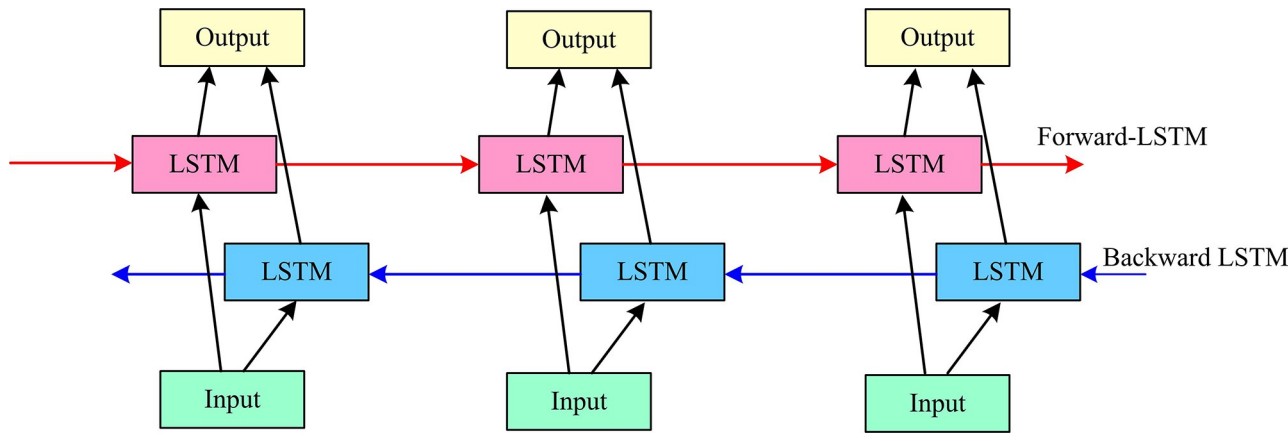

**Fig 2. BiLSTM network structure.**

respectively. $x_t$ is the input, and $b$ is the offset term. $h_t$ is the output. is vector splicing operation. Combined with the above, IBLSTM structure is shown in Fig 3.

In Fig 3, the current input is first interacted with the previous input to improve the contextual connection capability of the LSTM. BiLSTM is used for bidirectional interaction to achieve integrated input and output and increase the network performance.

## 3.2. IABLN recognition algorithm based on IBLSTM network

In order to realize the design of the IABLN algorithm, an attention mechanism is introduced to IBLSTM output attention weighting. The attention mechanism is to find the correlation between the data on the basis of the original data, and introduce a weight score according to the correlation to measure the output [27,28]. Attention mechanisms are mainly divided into soft and hard attention, among which hard attention cannot be differentiated in deep learning technology, so the simple soft attention mechanism is used to de-output weighting. The principle of the soft attention mechanism can be summarized as weighting different parts by assigning different weights to the existing input information. This weighting process improves model performance by allowing the model to focus more on information relevant to the task

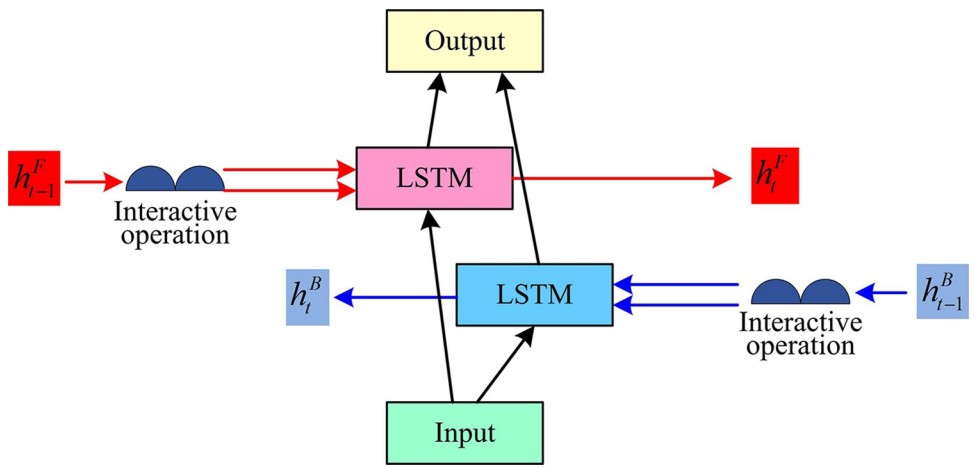

**Fig 3. IBLSTM network structure.**

at hand. In contrast to traditional hard attention mechanisms, soft attention mechanisms model the attention weight distribution as a probability distribution, thereby enabling the model to generate attention for all input positions, rather than just one location [29,30]. The specific expression is shown in Eq (3).

$$\text{Attn}(X, q) = \sum_{i=1}^{N} \alpha_i x_i \tag{3}$$

In Eq (3), Attn($\bullet$) represents the output of the sequence after being weighted by attention, $N$ represents input length, $X$ represents input length $N$ sequence, q represents the input query vector, $x_i$ represents the $i$ element in the sequence, and $\alpha_i$ represents the weighted value. The weighted formula for the attention mechanism is shown in Eq (4) [31].

$$\alpha_i = p(z = i | X, q) = \text{Softmax}(s(x_i, q)) \tag{4}$$

In Eq (4), $p$ represents the weighted probability, $z$ is selected information index position, and $s(\bullet)$ is the scoring function, which is calculated as shown in Eq (5) [32].

$$s(x_i, q) = x_i^{\text{T}} q \tag{5}$$

In Eq (5), T represents the transpose matrix. In order to more effectively illustrate the implementation steps of the soft attention mechanism, this study combines with the German translation of English task to describe it. Fig 4 shows the details.

In Fig 4, when translating the word "machine", it is first calculated by the attention mechanism to obtain an input. The current input represents the input of the current moment, which is used to calculate the score based on the hidden state of the previous unit and the output of each unit of the encoder. The current moment input is a weighted average of the score and the output of the encoder, as well as the output of the previous moment. The specific function

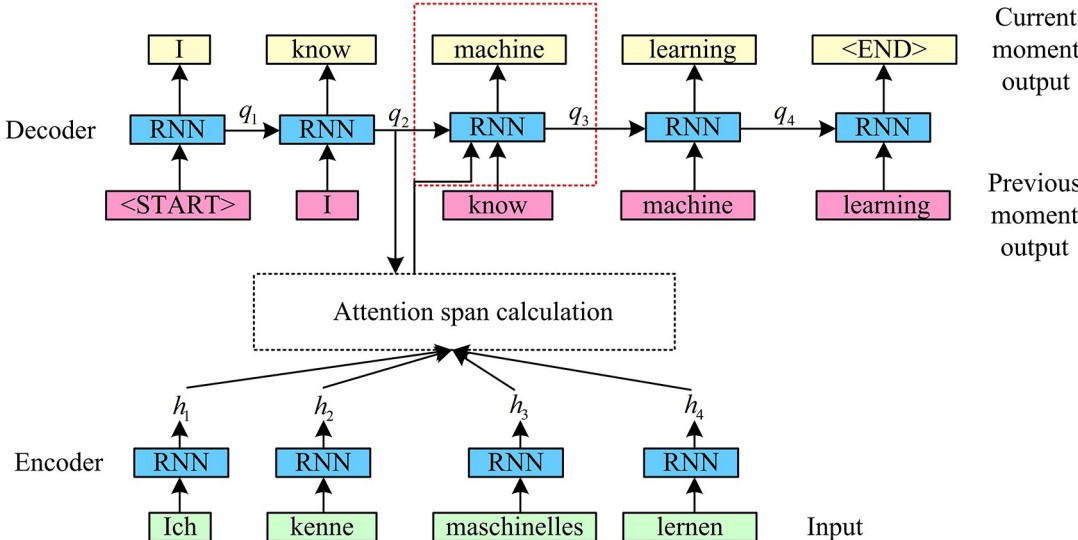

**Fig 4. Steps in the execution of translation tasks incorporating soft attention mechanisms.**

expression is shown in Eq (6).

$$\begin{cases} [\alpha_1, \alpha_2, \alpha_3, \alpha_4] = \text{Softmax}[s(q_2, h_1), s(q_2, h_2), s(q_2, h_3), s(q_2, h_4)] \\ \text{context} = \sum_{i=1}^{4} \alpha_i h_i \end{cases} \tag{6}$$

In Eq (6), context represents the input at this moment. The two inputs of the attention mechanism are both from the output of the BiLSTM, which is conducive to increasing the influence of the output in the LSTM unit on the modulation pattern recognition results to a certain extent, so as to improve network performance. Combined with the above information, the network structure of the IABLN algorithm based on IBLSTM is shown in Fig 5.

In Fig 5, the input data is the modulation data of the original automatic wireless communication system. The input signal is a non-inverting quadrature decomposition sequence signal with a length of 128 and a length of 128 and a number of channels of 2. The input channel of the convolutional layer is 2, the output channel is 128, and convolution kernel size is 5. The output of the convolutional kernel is activated and sent to the pooling layer, and the size of the pooling layer kernel and step size is set to 3, and pooling layer input is filled with an edge of 0. The time series network is a time series network containing 42 bidirectional interactive time series units, and the input dimension and hidden dimension are 128. Therefore, the flow of the modulation pattern recognition method based on the IABLN algorithm proposed in the study is shown in Fig 6.

In Fig 6, IABLN algorithm first passes the input signal through a one-dimensional convolutional layer for feature extraction, and expands the signal from two dimensions to a high-dimensionality. The down-sampled signal is transmitted to BiLSTM for temporal feature extraction, and the soft attention mechanism is used to weight the output. Finally, the output result is transmitted to the classification network for classification, and the modulation class output of the modulated signal is carried out. The forward and backward output hidden states of the BiLSTM are stitched together and used as hidden information of the attention layer, so

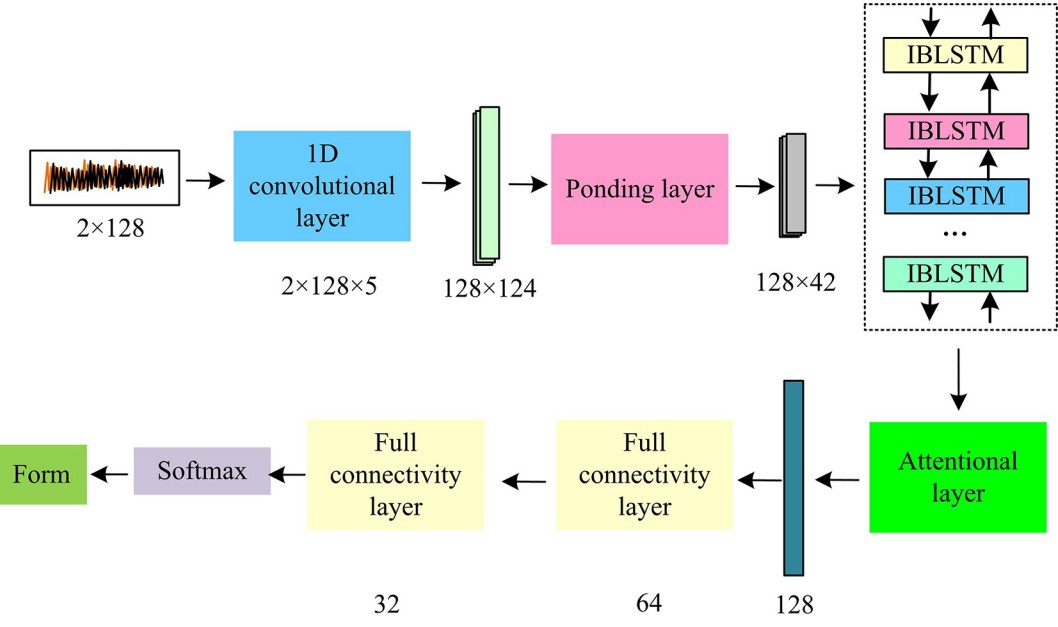

**Fig 5. Schematic diagram of the network structure of the IABLN recognition algorithm.**

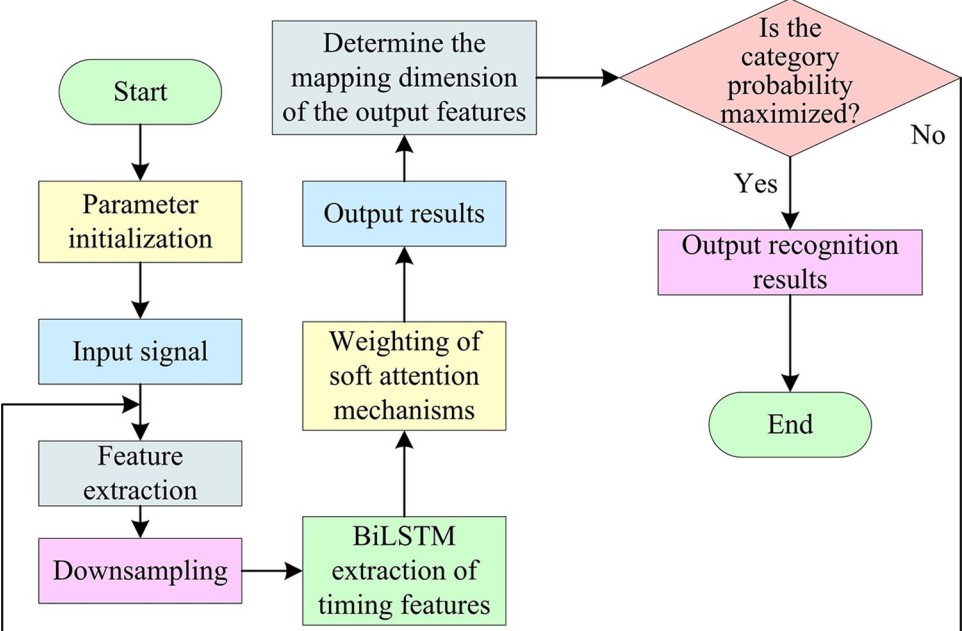

**Fig 6. Schematic diagram of the network structure of the IABLN recognition algorithm.**

as to weight the output of the time series network layer. Finally, through the fully connected layer of two layers, the mapping dimension of the output features is determined according to the number of modulation categories. After obtaining the class judgment probability of the signal through the Softmax layer, the category with the largest probability value is selected as the final modulation class judgment result. All outputs of the convolutional and fully connected layers are fitted to the nonlinear model through a network of excitation functions.

The IABLN algorithm effectively exploits the temporal properties of modulated signals by combining the BiLSTM and the attention mechanism.The bidirectional structure of the BiLSTM allows the algorithm to take into account both the past and the future information of the signal, which is crucial for understanding the long-term dependence of the signal. The attention mechanism further enhances the model's ability to identify critical time segments in the signal, improving the accuracy of modulation identification. At the same time, this study, the Parametric Rectified Linear Unit (PReLU) is the activation function [33]. The main expression is shown in Eq (7).

$$f_{\mathrm{PReLU}}(y_i) = \begin{cases} y_i & y_i \geq 0 \\ a_i y_i & y_i < 0 \end{cases} \tag{7}$$

In Eq (7), $f_{\mathrm{PReLU}}$ represents the PreLU excitation activation function, $y_i$ is nonlinear activation function input, and $a_i$ represents negative half axis slope. PReLU can excite all the output characteristics of the convolutional computational layer in a nonlinear manner. Without adding additional parameters, it can improve the over-fitting ability of the module and reduce the probability of overfitting. When negative half axis slope is controlled to 0, PreLU is transformed into a Rectified Linear Unit (ReLU) [34,35]. ReLU is a commonly used activation function, usually referring to the ramp function, while ReLU is aReLU with parameters. In addition, when the model is trained, the cross-entropy loss function calculates the loss [36,37].

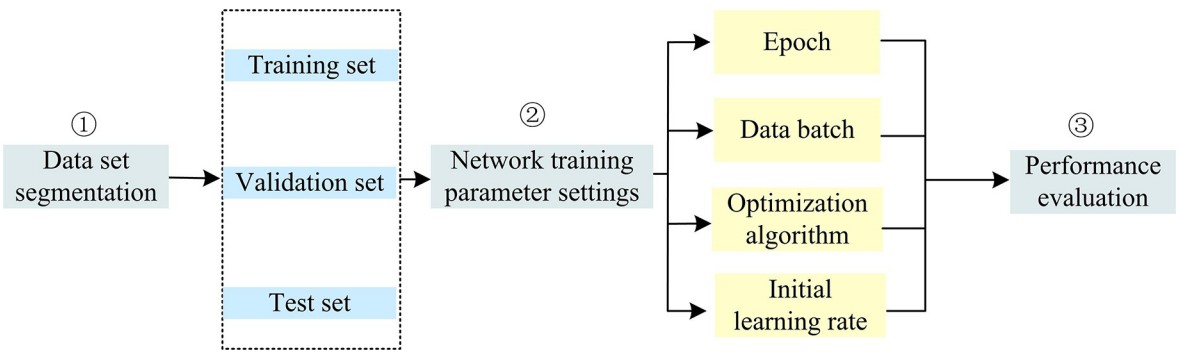

**Fig 7. Experimental steps of modulation pattern recognition method based on IABLN algorithm.**

The specific expression is shown in Eq (8).

$$\text{Loss} = -\frac{1}{m}\sum_{j=1}^{m}\sum_{i=1}^{n} y_{ji}\log(\hat{y}_{ji}) \tag{8}$$

In Eq (8), Loss is the cross-entropy loss function, $m$ is the amount of data input in a batch, $n$ is the number of types, and $y_{ji}$ is the probability that a sample label is a certain kind, with a value of 1 or 0. $\hat{y}_{ji}$ represents the probability that the network predicts that the sample will be a certain type of modulation. Finally, to verify the modulation pattern recognition method using proposed IABLN algorithm, a simulation experiment is designed. The whole phase consists of three modules, namely dataset division, network training parameter configuration, and network performance evaluation.

In Fig 7, a dataset consisting of multiple sampled modulation signals and corresponding modulation type labels is first classified. A plurality of data is randomly selected as the training data of each class under each SNR of a single modulation species, and multiple samples are randomly selected from the remaining data of a single SNR as the validation set. The last remaining sample is used as a testing machine. The training, validation, and test datasets of each class are combined into a training set, validation set, and testing machine, respectively. The K-fold cross-test is used to divide all data except the independent test set into K-part experiments that did not overlap each other [38,39]. Each K-1 copy is selected as the training set, and another one is used as the test set. The test sets selected for each time of the K experiments are not overlapped with each other, and the average value is taken as the experimental results. Since the types of each modulation mode are consistent with the number of samples, a non-stratified K-fold cross-test is used, and the K-value is set to 5. Secondly, the network training parameters are configured. It mainly includes the number of iterations, the size of the data batch sent to the network parameter training in the training batch, and the initial learning rate. In this study, the Adam optimizer algorithm is used as the optimization algorithm of the model, and the early stop strategy controls the loss process of the verification set. Meanwhile, the stochastic gradient descent optimizer with momentum is adjusted by the cosine annealing algorithm, and the initial learning rate is 0.001. The training is stopped when the validation set loss function exceeds ten generations and does not degrade the process. Finally, the network performance is evaluated. After the network finishes training, the effectiveness of this modulation pattern recognition method is tested by an independent detection set. The Overall Accuracy(OA), Average Accuracy (AA), Max Accuracy (MA), and Kappa Coefficient (KC) are

mainly used, wherein the OA calculation is shown in Eq (9) [40–42].

$$OA = \frac{\sum_{i=1}^{N_{class}} T_{ij}}{\sum_{i=1}^{N_{class}} C_i} \tag{9}$$

In Eq (9), $N_{class}$ is the number of categories, $T_{ij}$ is the number of signals identified as a certain class, and $C_i$ is the total samples number of a certain category of the testing machine. AA is calculated as shown in Eq (10).

$$AA = \frac{1}{N_{class}} \sum_{i=1}^{N_{class}} \frac{T_{ij}}{C_i} \tag{10}$$

KC is a quantitative expression of the confusion matrix, which indicates consistency degree between the predicted and actual results, and the closer the KC is to 1, the closer the predicted results are to actual results. The specific calculation is shown in Eq (11).

$$Kappa = \frac{OA - \frac{\sum_{i=1}^{N_{class}} C_i \times N_i}{N \times N}}{1 - \frac{\sum_{i=1}^{N_{class}} C_i \times N_i}{N \times N}} \tag{11}$$

In Eq (11), Kappa represents the KC.

## 4. Verification and analysis of modulation pattern recognition method using IABLN algorithm

To verify the modulation pattern recognition method using IABLN algorithm, performance verification of IBLSTM network based on LSTM was carried out firstly. Secondly, attention mechanism's effectiveness was verified. Finally, performance analysis of IABLN algorithm and the comparison of modulation pattern recognition methods were carried out.

### 4.1. LSTM-based IBLSTM network validation

In order to validate the effectiveness of the IBLSTM network proposed by the study, the study was conducted in RML 2016.10b to validate the effectiveness and analyze the effect of the number of interaction rounds on the performance. The RML2016.10b dataset had about 1.2 million data samples, and the modulation types mainly contained the eight most commonly used digital modulations and two analog modulation information. At the same time, one layer LSTM, two layer LSTM, bidirectional LSTM and two layer Gate Recurrent Unit (GRU), a variant of LSTM, were introduced for performance comparison. The five-fold cross-validation took the optimal weight of each model in 5 rounds to perform 5 inferences on the independent test set, and the average value was taken. It was worth mentioning that all validation experiments in the study were trained tests in a Linux runtime server consisting of an Intel i7 7800 processor and two Nvidia Titan Xp graphics processors. The deep learning framework was Pytorch 1.10 in a CUDA 11.5 environment and the runtime program was Python 3.8. The test results of the test set after the network used different time series networks to train the training set convergence, as shown in Fig 8.

In Fig 8A, the accuracy of the proposed IBLSTM time series network was the highest under each SNR. Compared with other time series networks, its recognition accuracy was superior. Among them, the SNR [–12,–2] interval had the most obvious increasing effect. When the

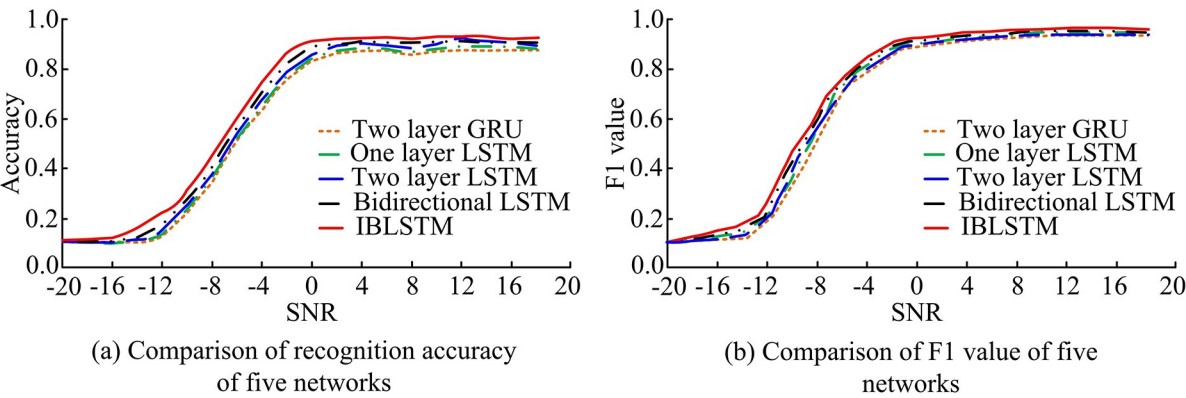

**Fig 8. Comparison of recognition performance using different temporal network networks at different signal-to-noise ratios.**

SNR value was 18dB, the proposed IBLSTM time series network increased by 2.34% on average compared with other methods. Fig 8B shows a comparison of F1 values for different time-series networks. It can be concluded that the overall difference of F1 values of the five time series networks was small, but IBLSTM still had superiority. When the SNR was 18dB, the F1 value of IBLSTM was as high as 0.95. Table 1 shows the overall performance evaluation results of the five time series networks.

In Table 1, OA of the proposed IBLSTM network was 63.92%, which was the highest among the five time series networks. OA represents the percentage between the number of prediction pairs and the overall amount of data, and the higher the value, the more accurate the overall recognition rate. AA represents an average of the recognition accuracy of each category. The proposed IBLSTM has the highest AA value. Compared with the two-layer GRU, the AA value of IBLSTM increased by 10.47%. Compared to standard LSTMs, IBLSTMs increased by an average of 7.39%. This indicated that the recognition performance of IBLSTM was superior. The MA value of IBLSTM was 92.80%, which was 1.87% higher than that of bidirectional LSTM. In the comparison of KCs, IBLSTM still showed better recognition results. On the whole, compared with other time-series networks, the proposed IBLSTM had a significant improvement in the recognition performance of modulation modes, which confirmed the feasibility and effectiveness of the improved LSTM.

## 4.2. Modulation pattern recognition performance verification based on IABLN algorithm

To verify the rationality and superiority of IABLN algorithm in modulation pattern recognition, the attention mechanism was first verified. The IBLSTM without the output weighting of the attention mechanism was employed as a control. The same weighting was applied to add the output content of all units, which was then input into the subsequent classification sub-

**Table 1. Comparison of indicator evaluation results for different time series networks.**

| Performance index | Temporal network | | | | |
|---|---|---|---|---|---|
| | Two layer GRU | One layer LSTM | Two layer LSTM | Bidirectional LSTM | IBLSTM |
| OA | 57.23% | 58.23% | 59.74% | 62.01% | 63.92% |
| AA | 58.45% | 59.38% | 60.90% | 62.07% | 64.57% |
| BUT | 87.60% | 89.09% | 91.97% | 91.10% | 92.80% |
| KC | 0.54 | 0.58 | 0.56 | 0.60 | 0.62 |

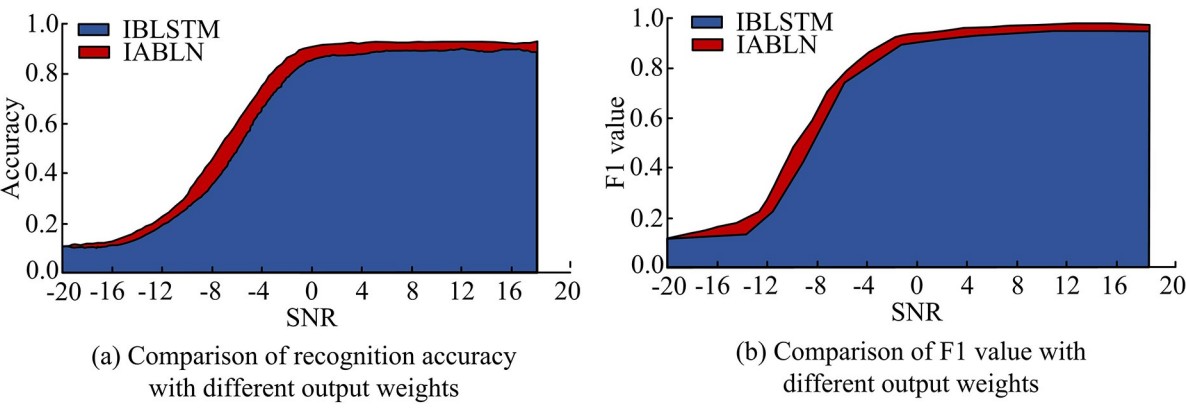

(a) Comparison of recognition accuracy
with different output weights

(b) Comparison of F1 value with
different output weights

**Fig 9. The recognition accuracy of various signal-to-noise ratios with and without attention mechanism in the network.**

network. All the remaining structures and positions of the rest of the networks remained unchanged. After five-fold cross-training, each chose the weight with the best effect on the validation set and tested it in the independent test set. Fig 9 shows the overall recognition accuracy and F1 value comparison results of the two weighting methods.

Fig 9A shows the recognition accuracy of the temporal network under different SNRs with and without attention mechanism, and it can be concluded that the accuracy of the temporal network after adding the attention mechanism was improved faster. When the SNR was -2dB, the recognition accuracy tended to be stable. This indicated that the convergence effect of the temporal network was further improved after the attention mechanism was added. Comparing Fig 9B, the F1 value of IABLN increased by 3.26% compared to IBLSTM. This showed that the attention mechanism effectively improved modulation pattern recognition accuracy in the network. Fig 10 shows the performance indicators of the two methods.

In Fig 10A and 10B, the performance evaluation results of the IBLSTM network without attention weighting were worse than those with attention mechanism weighting. After adding the attention mechanism, OA increased by 10.34%, AA increased by 8.33%, and MA increased by 3.33%. In addition, the KC of the IABLN network after adding the attention mechanism

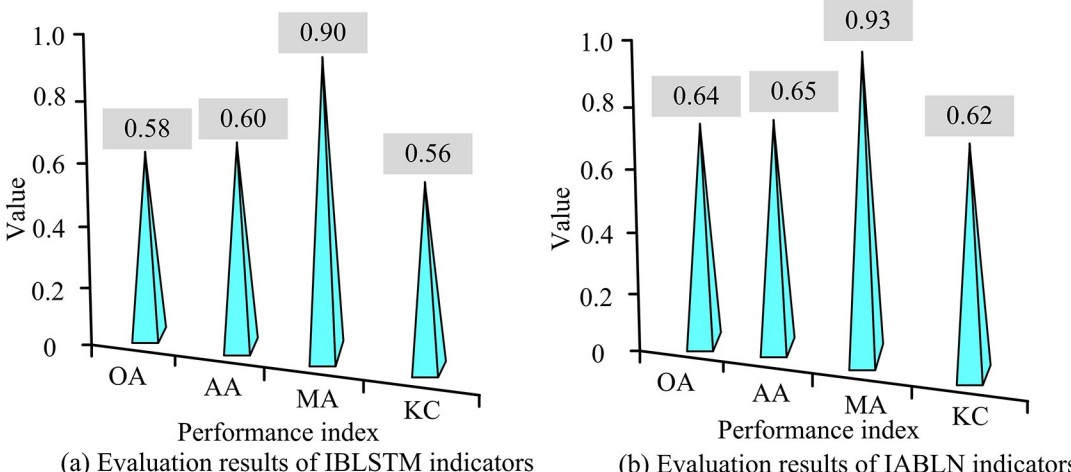

(a) Evaluation results of IBLSTM indicators

(b) Evaluation results of IABLN indicators

**Fig 10. Performance evaluation results of network with and without attention mechanism.**

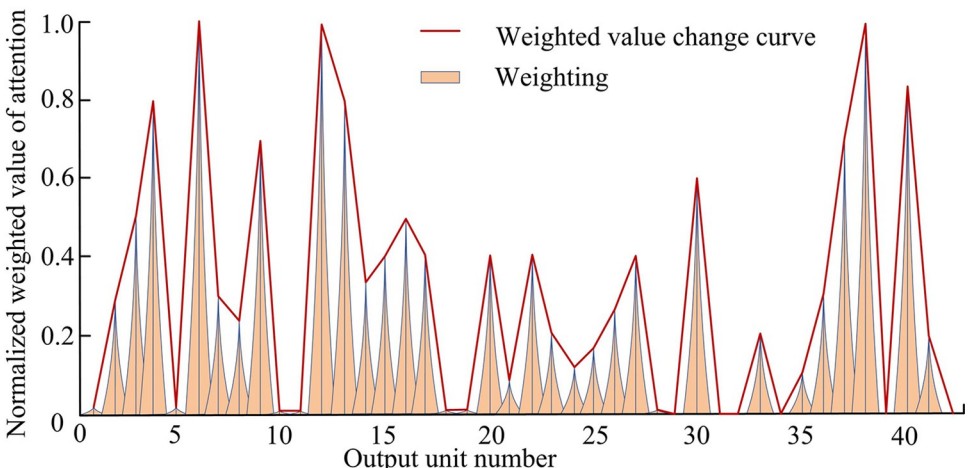

**Fig 11. The weighting of attention mechanism for each output unit when inputting a 14 dB PAM4 signal.**

was 0.06 higher than that of IBLSTM. The above results showed that the addition of attention mechanism effectively improved the recognition effect of the network on the modulation pattern and enhanced temporal network performance. To better understand the enhancement effect of the attention mechanism, the output score generated by the attention unit in the PAM4 modulated signal with an input SNR of 14dB in the RML2016 dataset was analyzed. The normalized weighted results are shown in Fig 11.

In Fig 11, the attention mechanism differed in the weighting of the output of each bidirectional interactive time series unit, indicating that the output at some locations was filtered and trade-off. The weighted values of output units 6, 12 and 38 were as high as 1.00, which indicated that the attention mechanism attached more importance to these three elements, and also indicated that the output results of these three elements had a greater influence on the whole modulation pattern recognition results. Finally, to further reflect the effectiveness of the proposed IABLN algorithm for modulation pattern recognition, the test set was verified by using the best network weights in the verification set. Fig 12 shows the modulation pattern recognition results of IABLN algorithm with SNR of -12dB and 0dB in the test set.

In Fig 12A, confusion matrix diagonal value was lower with SNR of -12 dB, indicating that the recognition accuracy was lower at -12 dB. Compared with Fig 12B, the recognition accuracy of IABLN algorithm was improved with SNR increased. Among the 10 kinds of modulated signals, the recognition accuracy of the CPFSK modulated signal was 1.00. Among them, that of WBFM modulated signal was the lowest, which was only 0.32. Therefore, the confusion matrix of the IABLN algorithm was further analyzed when the SNR of the test set was 6dB and 10dB, as shown in Fig 13.

In Fig 13A, the vast majority of modulation mode signals could be accurately identified at an SNR of 6 dB. Among them, the recognition accuracy of CPFSK, GFSK and PAM4 modulated signals was as high as 1.00. Fig 13B shows the recognition results at a SNR of 10 dB, and the recognition effect of the modulated mode signal was superior when the SNR was 10 dB compared to the first three SNRs. However, there was a certain misjudgment in the identification of QAM-like signals, which may be due to the fact that the modulated signal QAM16 was input into a subset of QAM64 during the modulation process, causing the IABLN algorithm to classify it as a QAM64 modulated signal. The modulation recognition accuracy of WBFM modulated signals was the lowest under different SNRs, indicating that the relationship between this mode and SNR was small. This discrepancy may be attributed to the fact that

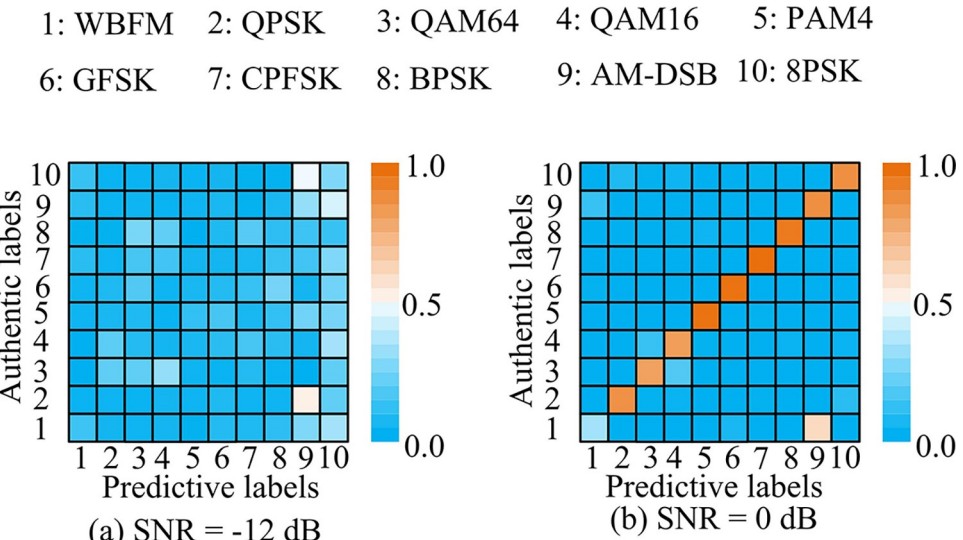

**Fig 12. Modulation pattern recognition results with SNR of -12dB and 0dB in the test set.**

WBFM modulated signals, like AM-DSB modulated signals, were analog modulated. This process involved a continuous speech signal as a data source, whose model was affected by the silence time period. Additionally, the different SNRs differentiated them to a lesser extent, which results in a significantly lower recognition accuracy of the IABLN for such signals in comparison to digitally modulated signals. Most of the modulation modes increased as SNR increased, and QAM16 as the most affected by SNR.

## 4.3 Performance validation of different methods

To further illustrate the superiority of the IABLN algorithm, the study introduces the current commonly used modulation pattern recognition methods, as well as references [12,18], for

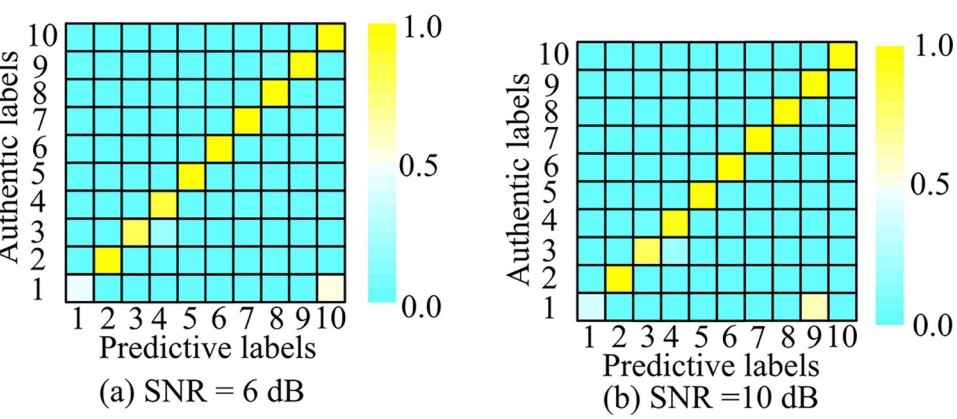

**Fig 13. Modulation pattern recognition results with SNR of 6dB and 10dB in the test set.**

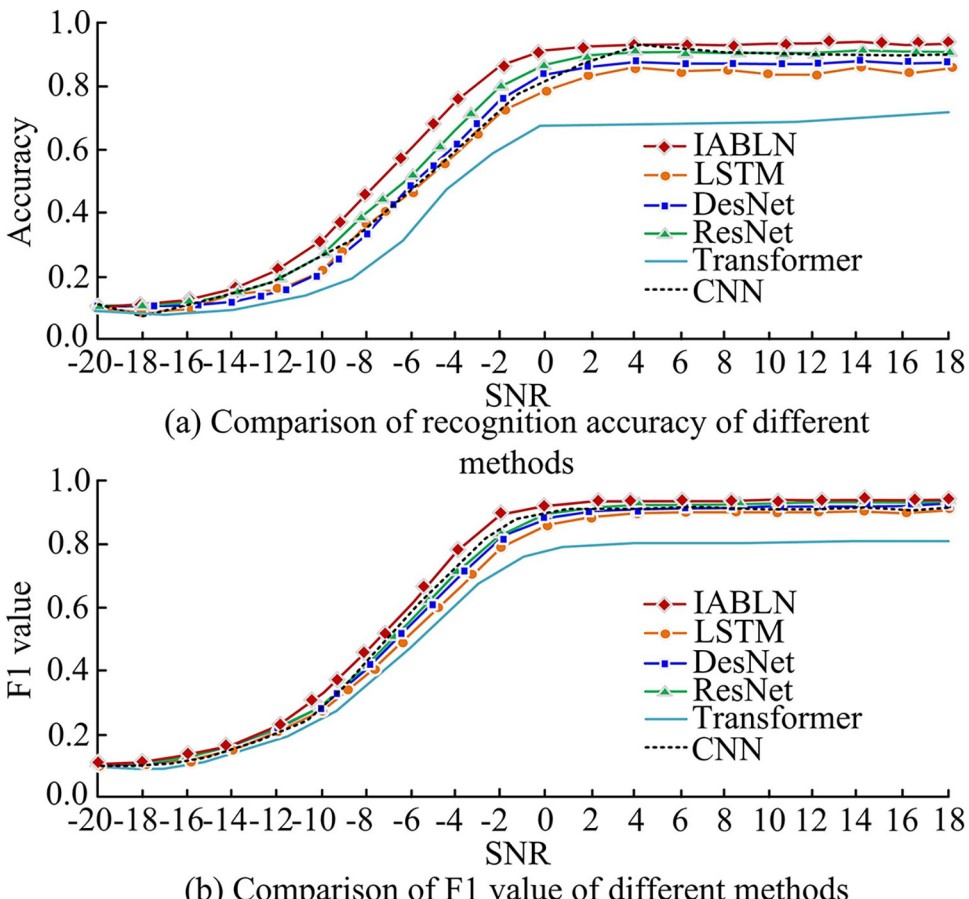

(a) Comparison of recognition accuracy of different methods

(b) Comparison of F1 value of different methods

**Fig 14. Recognition results of IABLN and comparative methods on the test set under different SNRs.**

comparison with IABLN. Among them, the model proposed in literature [12] is Transformer and the model proposed in literature [18] is CNN. Meanwhile, the study employs the optimal parameter settings identified by the literature's authors as the relevant parameter settings for the comparison methods. First, the study compares the recognition accuracy and F1 value of the six methods in the RML2016 10.b dataset. The specific comparison results are shown in Fig 14.

Comparing the recognition accuracy of the four recognition methods in Fig 14A, it can be concluded that when the SNR was -2dB, the frequency of the recognition accuracy increase of the four methods gradually tended to be stable. The recognition accuracy of the proposed recognition method was improved the fastest. When SNR was greater than 6dB, the highest recognition accuracy of IABLN reached 92.84%. In Fig 14B, F1 of LSTM was the worst among the four methods, while the F1 value of the IABLN proposed in the study was 3.33% higher than that of the LSTM. Meanwhile, the study contributes to the evaluation of the six methods on different hardware, providing insight into the parameters of the model and the inference speed. The specific results are presented in Table 2.

Table 2 demonstrates that the modulation recognition method based on IABLN exhibits enhanced recognition accuracy compared to existing methods. The total number of parameters of IABLN was as high as 300,000, which afforded greater computing power than other methods. In terms of inference time, the response speed of IABLN was 1.57 ms, which was

**Table 2. Performance evaluation results of IABLN and comparative algorithms.**

| Performance index | Method | | | | | |
|---|---|---|---|---|---|---|
| | LSTM | DesNet | ResNet | IABLN | Transformer | CNN |
| Total number of parameters | 220.00k | 300.00k | 200.00k | 300.00k | 250.00k | 240.00k |
| Inference time | 22.33ms | 24.12ms | 17.88ms | 1.57ms | 2.45ms | 17.88ms |
| OA | 52.10% | 55.76% | 60.34% | 62.88% | 57.89% | 60.02% |
| AA | 54.34% | 57.44% | 61.33% | 63.89% | 55.28% | 59.64% |
| KC | 0.52 | 0.54 | 0.60 | 0.63 | 0.55 | 0.58 |

90.73% lower than that of other methods on average. In comparison to the OA value of LSTM, the recognition accuracy of IABLN exhibited a notable enhancement. In comparison to Des-Net and ResNet, the OA value of IABLN exhibited an increase of 12.77% and 4.21%, respectively. The KC of IABLN was 0.63, representing an average increase of 12.90% compared with other methods. Although ResNet had a smaller number of parameters than the IABLN algorithm, it was less efficient in terms of inference time. This may be due to the IABLN algorithm's ability to be deployed in an offline device after it has been trained on appropriate data, enabling real-time responses. The preceding validation results demonstrated that the research proposal, IABLN, exhibits notable advantages in the RML2016.10b dataset.

On this basis, the study further utilized the CSPB.ML2018 dataset and the RML2016.09a dataset for performance checking. The CSPB.ML2018 dataset was an improvement of the RML2016 10.b dataset, which addressed the known issues and survey errors of the RML2016 dataset. It contained 8 different digital modulation modes, 3584000 signal samples. The RML2016.09a dataset was an earlier version of the RML2016 dataset that also contained signal samples for multiple modulation types. The performance evaluation results of the six methods on the CSPB.ML2018 dataset are shown in Table 3.

Table 3 indicates that the proposed IABLN algorithm of the study exhibits reduced inference time and recognition accuracy relative to Transformer, although the discrepancy was marginal. Nevertheless, the inference time of the IABLN algorithm was reduced by an average of 79.63% in comparison to DesNet, ResNet, and CNN. This discrepancy may be attributed to the fact that Transformer optimized the design of the model in terms of size and performance during the experimental phase, thereby conferring an advantage to its proposed Transformer architecture in the identification of signal modulation in the CSPB.ML2018 dataset.

**Table 3. Performance evaluation results of the six methods in the CSPB.ML2018 dataset and RML2016.09a dataset.**

| Method and datasets | | Total number of parameters | Inference time | OA | AA | KC | *p*-value |
|---|---|---|---|---|---|---|---|
| LSTM | CSPB.ML2018 | 220.00k | 30.24ms | 50.55% | 51.02% | 0.5 | 0.001 |
| | RML2016.09a | 200.00k | 19.87ms | 65.55% | 54.65% | 0.46 | |
| DesNet | CSPB.ML2018 | 300.00k | 28.56ms | 56.87% | 55.46% | 0.53 | 0.001 |
| | RML2016.09a | 250.00k | 20.09ms | 61.02% | 60.88% | 0.5 | |
| ResNet | CSPB.ML2018 | 200.00k | 21.88ms | 64.72% | 63.25% | 0.61 | 0.001 |
| | RML2016.09a | 180.00k | 15.68ms | 70.75% | 68.48% | 0.64 | |
| IABLN | CSPB.ML2018 | 300.00k | 4.95ms | 65.07% | 64.21% | 0.62 | - |
| | RML2016.09a | 280.00k | 3.86ms | 78.65% | 74.21% | 0.68 | |
| Transformer | CSPB.ML2018 | 250.00k | 4.54ms | 65.60% | 64.33% | 0.64 | 0.098 |
| | RML2016.09a | 200.00k | 4.01ms | 80.09% | 75.51% | 0.67 | |
| CNN | CSPB.ML2018 | 240.00k | 22.45ms | 59.78% | 56.54% | 0.53 | 0.003 |
| | RML2016.09a | 200.00k | 12.33ms | 68.78% | 70.23% | 0.62 | |

Nevertheless, the modulation pattern recognition method based on the IABLN algorithm proposed in the study remained demonstrably superior in terms of recognition accuracy and recognition efficiency. Comparing the validation results of different algorithms on the RML2016.09a dataset, it can be concluded that the proposed method of the study was more superior in terms of response time. Furthermore, the recognition accuracy of modulation patterns can be significantly enhanced by the utilization of the IABLN algorithm. The superiority of the study can also be illustrated by comparing the test of significance obtained by repeating the test 3 times with different methods ($P<0.05$).

## 4.4 Time complexity analysis

Finally, the study further performs a time complexity analysis. Assuming that the length of the input signal was N, the size of the convolutional kernel was K, the number of input channels was $C_{in}$, and the number of output channels was $C_{out}$. The computational complexity of each output feature map was $O(N \cdot C_{in} \cdot K)$, and the total complexity of the convolutional layer was $O(C_{out} \cdot N \cdot C_{in} \cdot K)$. For each direction of the LSTM, the time complexity was $O(N \cdot D)$. Among them, D was the hidden layer dimension of the LSTM cell. Since the LSTM used in the study was bi-directional, the total time complexity of BiLSTM was $O(2 \cdot N \cdot D)$. The attentional mechanism necessitates the scoring and normalization of each element within the sequence, which was of a complexity of $O(N \cdot D)$ for each output time step and a total time complexity of $O(N^2 \cdot D)$. Assuming that the output dimension of the fully connected layer was M, the complexity was $O(N \cdot D \cdot M)$. Combining the above, the total time complexity of the IABLN algorithm proposed in the study was $O(N \cdot (C_{out} \cdot C_{in} \cdot K + 3 \cdot D + D \cdot M) + N^2 \cdot D)$.

The computational complexity of the traditional one-way LSTM was less than that of the IABLN, as it was only processed once at each time step. Conversely, CNNs primarily extract features through convolutional and pooling layers, which typically exhibit a reduced computational complexity, particularly when processing high-dimensional data such as images. However, CNNs do not typically address temporal information. The Transformer deals with the data by means of a self-attentive mechanism, which can process all elements in a sequence in parallel. Moreover, the computational complexity was related to the length of the sequence and the number of attention heads. Although Transformer performed well in some tasks, its computational complexity was usually higher than that of LSTM, especially when the sequence length was long. The preceding experimental results demonstrated that the IABLN algorithm exhibited high recognition accuracy and a rapid response time on diverse datasets. These findings suggested that the algorithm was an effective approach for modulation recognition tasks, despite its high computational complexity.

## 5. Discussion

The study proposed a signal modulation recognition method based on the IABLN algorithm and validated its performance on the RML2016 and CSPB.ML2018 datasets. The IBLSTM network was trained on the RML2016 dataset in the training set, and its test results exhibited an average increase of 2.34% compared to those of other networks. In their study, M. A. Hamza et al. employed the BiLSTM to investigate the modulated signals of communication systems through the lens of deep learning models. The outcomes of this investigation aligned with the findings of the aforementioned study. This indicated the rationality and feasibility of using BiLSTM to enhance the contextual connection of temporal networks. The introduction of the BiLSTM network, coupled with the use of an attention mechanism to weight the network output, had been demonstrated to enhance the network's recognition efficiency. This was corroborated by the study of M. Tian et al. [43]. Therefore, it can be concluded that the modulation

pattern recognition method for wireless communication automated systems based on the IABLN algorithm proposed in the study was both reasonable and effective.

The results of the performance comparison of various methods on the RML2016 and CSPB. ML2018 datasets indicated that the number of parameters of the different methods was not consistently correlated with the actual inference time. In comparison to LSTM, ResNet, Reference [12], and Reference [18], the IABLN algorithm exhibits a greater number of parameters. However, it demonstrated a notable advantage in inference time. Y. Zang et al. conducted a study on data-driven fiber optic models using BiLSTM with an attention mechanism. Their findings indicated that, with a high number of parameters and a complex modulation format, the proposed model exhibits a faster prediction speed. This indicated that strengthening the temporal network contextual links enhances the recognition speed of the IABLN, while the soft attention mechanism weights the output of the temporal network, which in turn accelerates the algorithm training process, thereby improving the overall performance of the modulation pattern recognition method.

It was important to note that both the RML2016 and CSPB.ML2018 datasets utilized in the study were heavily labeled types of data. Deep learning algorithms were generally reliant on the sample size of the data, and the limited training data may result in a degradation of the algorithm's performance. In light of the limited data volume in practical applications, the study proposed the introduction of data augmentation to the original data in subsequent work. This approach extended the data volume by adding noise or generating new samples, thereby enabling the algorithm to perform more effectively. With regard to the issue of an imbalanced distribution of actual modulation patterns, the study proposed the introduction of machine learning algorithms for the enhancement and optimization of the IABLN algorithm in subsequent work. This could the identification and classification of imbalanced modulation patterns with multiple models or variants. Furthermore, due to the challenging nature of acquiring models with labels in real-world environments, the study primarily conducted performance validation on publicly available datasets. In light of the intricate nature of automatic modulation recognition, it was evident that there were numerous avenues for further research and optimization. The research shortcomings were presented in order to facilitate the advancement of wireless communication technology, which will benefit human society. This was achieved through an in-depth exploration of signal modulation identification methods.

## 6. Conclusion

To improve the accuracy of modulation pattern recognition in automatic wireless communication systems, a signal modulation recognition method using IABLN algorithm was designed. Firstly, BiLSTM was introduced on the basis of LSTM to enhance the context connection of the time series network, and the convolutional layer fit input signal features to reduce the length of the input stream, so as to construct the IBLSTM time series network. Finally, the soft attention mechanism was used to weight the output of the temporal network. Experimental verification on the RML2016 of the modulation dataset showed that the recognition accuracy of the proposed IBLSTM time series network was increased by 2.34% on average compared with other methods. After adding the attention mechanism, the F1 value of the time series network increased by 3.26%, OA increased by 10.34%, AA increased by 8.33%, and MA increased by 3.33%. Compared with other recognition methods, the IABLN recognition accuracy proposed in this study was as high as 92.84%, the response speed was only 2.45ms, and the KC increased by 12.90% on average. In the CSPB.ML2018 dataset, the IABLN algorithm inference also exhibited an average reduction of 7% in comparison to the other algorithms. Results showed that this proposed method using the IABLN algorithm had higher recognition

accuracy on each SNR, and the inference recognition speed was faster when the number of parameters was small, and the recognition performance was superior to other recognition methods. However, there are still some shortcomings in the research. The simulation experiments were mainly carried out by using a public dataset for verification, and the modulation signal experiments in the actual environment were not carried out. In the future, this study will further explore how to obtain more accurate data in the actual environment for the robustness verification of the recognition method, so as to improve the application value of the modulation pattern recognition method.

## Supporting information

**S1 Dataset. Minimal data set definition.**
(DOC)

## Author Contributions

**Conceptualization:** Ting Xie.

**Data curation:** Ting Xie.

**Funding acquisition:** Xing Han.

**Investigation:** Xing Han.

**Validation:** Xing Han.

**Writing – original draft:** Ting Xie.

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
