## [Decision Letter · Decision Letter 0]

15 Aug 2024

PONE-D-24-21497Modulation Pattern Recognition Method of Wireless Communication Automatic System Based on IABLN Algorithm in Intelligent SystemPLOS ONE

Dear Dr. Han,

Thank you for submitting your manuscript to PLOS ONE. After careful consideration, we feel that it has merit but does not fully meet PLOS ONE’s publication criteria as it currently stands. Therefore, we invite you to submit a revised version of the manuscript that addresses the points raised during the review process.

We look forward to receiving your revised manuscript.

Kind regards,

Salim Heddam

Academic Editor

PLOS ONE

Journal Requirements:

3. We note that your Data Availability Statement is currently as follows: "All relevant data are within the manuscript and its Supporting Information files."

4. We notice that your supplementary figures are included in the manuscript file. Please remove them and upload them with the file type 'Supporting Information'. Please ensure that each Supporting Information file has a legend listed in the manuscript after the references list.

Additional Editor Comments:

Reviewer 1#:（1）The added network model must be analyzed for performance evaluation.

（2）The contribution of this work must be clearly stated in the introduction section.

（3）In the last paragrapgh of section 1, the section number has a problem.

（4）The computational complexity of the proposed method should be analyzed to compared with the other representative methods.

（5）Use different datasets to verify your method.

Reviewer 2#:1 Clarify your writing, like IABLN in the first time, like the specific version of the dataset.

2 Focus on introduce and elaborate the interactive operation part further.

3 The comparison with state-of-the-art methods needs to be improved.

Reviewers' comments:

Reviewer's Responses to Questions

**Comments to the Author**

1. Is the manuscript technically sound, and do the data support the conclusions?

Reviewer #1: Partly

Reviewer #2: Partly

2. Has the statistical analysis been performed appropriately and rigorously? 

Reviewer #1: Yes

Reviewer #2: N/A

3. Have the authors made all data underlying the findings in their manuscript fully available?

Reviewer #1: Yes

Reviewer #2: Yes

4. Is the manuscript presented in an intelligible fashion and written in standard English?

Reviewer #1: Yes

Reviewer #2: No

5. Review Comments to the Author

Reviewer #1: （1）The added network model must be analyzed for performance evaluation.

（2）The contribution of this work must be clearly stated in the introduction section.

（3）In the last paragrapgh of section 1, the section number has a problem.

（4）The computational complexity of the proposed method should be analyzed to compared with the other representative methods.

（5）Use different datasets to verify your method.

Reviewer #2: 1 Clarify your writing, like IABLN in the first time, like the specific version of the dataset.

2 Focus on introduce and elaborate the interactive operation part further.

3 The comparison with state-of-the-art methods needs to be improved.

6. PLOS authors have the option to publish the peer review history of their article (what does this mean?). If published, this will include your full peer review and any attached files.

Reviewer #1: No

Reviewer #2: No

---

## [Author Response · Author response to Decision Letter 0]

2 Oct 2024

The manuscript has been revised according to comments.

---

## [Decision Letter · Decision Letter 1]

16 Oct 2024

PONE-D-24-21497R1Modulation Pattern Recognition Method of Wireless Communication Automatic System Based on IABLN Algorithm in Intelligent SystemPLOS ONE

Dear Dr. Han,

Thank you for submitting your manuscript to PLOS ONE. After careful consideration, we feel that it has merit but does not fully meet PLOS ONE’s publication criteria as it currently stands. Therefore, we invite you to submit a revised version of the manuscript that addresses the points raised during the review process. Please submit your revised manuscript by Nov 30 2024 11:59PM. If you will need more time than this to complete your revisions, please reply to this message or contact the journal office at plosone@plos.org. Please include the following items when submitting your revised manuscript:A rebuttal letter that responds to each point raised by the academic editor and reviewer(s). You should upload this letter as a separate file labeled 'Response to Reviewers'.A marked-up copy of your manuscript that highlights changes made to the original version. You should upload this as a separate file labeled 'Revised Manuscript with Track Changes'.An unmarked version of your revised paper without tracked changes. You should upload this as a separate file labeled 'Manuscript'.

We look forward to receiving your revised manuscript.

Kind regards,

Salim Heddam

Academic Editor

PLOS ONE

Additional Editor Comments:

Reviewer 1#:The authors have completed the paper revisions according to the reviewers' comments. It can be accepted for publication.

Reviewer 2#:Justify the rationals of the proposed method, i.e., what is the temporal characteristics of modulation data? (not to illustrate with voice data like in the paper).

Reviewers' comments:

Reviewer's Responses to Questions

**Comments to the Author**

1. If the authors have adequately addressed your comments raised in a previous round of review and you feel that this manuscript is now acceptable for publication, you may indicate that here to bypass the “Comments to the Author” section, enter your conflict of interest statement in the “Confidential to Editor” section, and submit your "Accept" recommendation.

Reviewer #1: All comments have been addressed

Reviewer #2: All comments have been addressed

2. Is the manuscript technically sound, and do the data support the conclusions?

Reviewer #1: Yes

Reviewer #2: Partly

3. Has the statistical analysis been performed appropriately and rigorously? 

Reviewer #1: Yes

Reviewer #2: Yes

4. Have the authors made all data underlying the findings in their manuscript fully available?

Reviewer #1: Yes

Reviewer #2: Yes

5. Is the manuscript presented in an intelligible fashion and written in standard English?

Reviewer #1: Yes

Reviewer #2: Yes

6. Review Comments to the Author

Reviewer #1: The authors have completed the paper revisions according to the reviewers' comments. It can be accepted for publication.

Reviewer #2: Justify the rationals of the proposed method, i.e., what is the temporal characteristics of modulation data? (not to illustrate with voice data like in the paper).

7. PLOS authors have the option to publish the peer review history of their article (what does this mean?). If published, this will include your full peer review and any attached files.

Reviewer #1: No

Reviewer #2: No

---

## [Author Response · Author response to Decision Letter 1]

10 Dec 2024

The manuscript has been modified according to comments.

Thank you very much!

---

## [Decision Letter · Decision Letter 2]

27 Dec 2024

Modulation Pattern Recognition Method of Wireless Communication Automatic System Based on IABLN Algorithm in Intelligent System

PONE-D-24-21497R2

Dear Dr. Han

We’re pleased to inform you that your manuscript has been judged scientifically suitable for publication and will be formally accepted for publication once it meets all outstanding technical requirements.

Kind regards,

Salim Heddam

Academic Editor

PLOS ONE

Additional Editor Comments (optional):

Reviewer 2#: A two-way interactive temporal network is designed on the basis of the long and short-term memory network with the objective of enhancing the contextual connection of the temporal network. The output of the temporal network is attentively weighted using the soft attention mechanism. The proposed algorithm exhibited enhanced overall, average, and maximum recognition rates at varying signal-to-noise ratios.

I suggest accept.

Reviewers' comments:

Reviewer's Responses to Questions

**Comments to the Author**

1. If the authors have adequately addressed your comments raised in a previous round of review and you feel that this manuscript is now acceptable for publication, you may indicate that here to bypass the “Comments to the Author” section, enter your conflict of interest statement in the “Confidential to Editor” section, and submit your "Accept" recommendation.

Reviewer #2: All comments have been addressed

2. Is the manuscript technically sound, and do the data support the conclusions?

Reviewer #2: Yes

3. Has the statistical analysis been performed appropriately and rigorously? 

Reviewer #2: Yes

4. Have the authors made all data underlying the findings in their manuscript fully available?

Reviewer #2: Yes

5. Is the manuscript presented in an intelligible fashion and written in standard English?

Reviewer #2: Yes

6. Review Comments to the Author

Reviewer #2: A two-way interactive temporal network is designed on the basis of the long and short-term memory network with the objective of enhancing the contextual connection of the temporal network. The output of the temporal network is attentively weighted using the soft attention mechanism. The proposed algorithm exhibited enhanced overall, average, and maximum recognition rates at varying signal-to-noise ratios.

I suggest accept.

7. PLOS authors have the option to publish the peer review history of their article (what does this mean?). If published, this will include your full peer review and any attached files.

Reviewer #2: No

---

## [Editor Report · Acceptance letter]

3 Jan 2025

PONE-D-24-21497R2 

PLOS ONE

Dear Dr. Han, 

I'm pleased to inform you that your manuscript has been deemed suitable for publication in PLOS ONE. Congratulations! Your manuscript is now being handed over to our production team.

Kind regards, 

on behalf of

Dr. Salim Heddam 

Academic Editor

PLOS ONE